# Prognostic Value of Tie2-Expressing Monocytes in Chronic Lymphocytic Leukemia Patients

**DOI:** 10.3390/cancers13112817

**Published:** 2021-06-05

**Authors:** Justyna Woś, Sylwia Chocholska, Wioleta Kowalska, Waldemar Tomczak, Agata Szymańska, Agnieszka Karczmarczyk, Agnieszka Szuster-Ciesielska, Agnieszka Wojciechowska, Agnieszka Bojarska-Junak

**Affiliations:** 1Chair and Department of Clinical Immunology, Medical University of Lublin, 20-093 Lublin, Poland; wioleta.kowalska@umlub.pl; 2Department of Haematooncology and Bone Marrow Transplantation, Medical University of Lublin, 20-080 Lublin, Poland; sylwia.chocholska@umlub.pl (S.C.); waldemar.tomczak@umlub.pl (W.T.); 3Department of Clinical Transplantology, Medical University of Lublin, 20-093 Lublin, Poland; agata.szymanska@umlub.pl; 4Department of Experimental Hematooncology, Medical University of Lublin, 20-093 Lublin, Poland; agnieszka.piechnik@umlub.pl; 5Department of Virology and Immunology, Institute of Biological Sciences, Maria Curie-Skłodowska University, 20-033 Lublin, Poland; szusterciesielska.agnieszka@poczta.umcs.lublin.pl; 6Faculty of Chemistry, Wroclaw University of Science and Technology, 50-370 Wrocław, Poland; agnieszka.wojciechowska@pwr.edu.pl

**Keywords:** Tie2, TEM, monocytes, CLL

## Abstract

**Simple Summary:**

Tie2-expressing monocytes (TEM) characterized by the phenotype of CD14+CD16+Tie2+ are seen as the new immunosuppressive force in tumors. However, little is known about the role of circulating TEM in chronic lymphocytic leukemia (CLL) as opposed to their role in solid tumors. In the current study, we observed an increased percentage of TEMs in CLL patients. A greater than 14.82% proportion of TEM foretells an unfavorable prognosis. This threshold has predicted a shorter time from diagnosis to therapy, and worse overall survival. Despite these results, a multivariable Cox regression model performed in 104 CLL patients did not identify TEM as an independent predictor of survival. However, TEM, as an important element of the tumor-microenvironment, can be an important complement to other prognostic indicators.

**Abstract:**

Tie2-expressing monocytes (TEMs) are associated with tumor progression and metastasis. This unique subset of monocytes has been identified as a potential prognostic marker in several solid tumors. However, TEMs remain poorly characterized in hematological cancers, including chronic lymphocytic leukemia (CLL). This study analyzed, for the first time, the clinical significance of TEM population in CLL patients. Flow cytometry analysis of TEMs (defined as CD14^+^CD16^+^Tie2^+^ cells) was performed at the time of diagnosis on peripheral blood mononuclear cells from 104 untreated CLL patients. Our results revealed an expansion of circulating TEM in CLL patients. These monocytes express high levels of VEGF and suppressive IL-10. A high percentage of TEM was associated closely with unfavorable prognostic markers (ZAP-70, CD38, 17p and 11q deletion, and IGHV mutational status). Moreover, increased percentages of circulating TEMs were significantly higher in patients not responding to the first-line therapy as compared to responding patients, suggesting its potential predictive value. High TEM percentage was also correlated with shorter overall survival (OS) and shorter time to treatment (TTT). Importantly, based on multivariate Cox regression analysis, TEM percentage was an independent predictor for TTT. Thus, we can suggest the adverse role of TEMs in CLL.

## 1. Introduction

Numerous studies have shown that peripheral blood monocytes are a heterogeneous cell population [1,2,3]. Some blood-circulating monocytes express the angiopoietin receptor Tie2 (Tie2-expressing monocytes, TEMs) [3]. Under physiological conditions, the TEM population represents 2 to 7% of blood mononuclear cells (about 20% of a monocytic population) [4,5]. However, the number of TEM can significantly increase in various pathological conditions, including cancer [4,6,7,8]. Angiogenesis stimulation is probably the key role of TEMs [3,5,9]. However, the molecular basis of the proangiogenic activity of this monocyte subpopulation is still not fully explained. It is known that the abundance and activity of Tie2-positive monocytes are directly dependent on angiopoietin-2 (Ang-2) [7,10,11]. TEMs migrate to Ang-2, released by activated endothelial cells and newly formed vessels within a tumor, which is suggested by a mechanism of TEMs targeting tumor tissue [4]. Interestingly, leukemic lymphocytes in CLL secrete Ang-2 [12]. Therefore, TEM function may also depend on CLL cells. It has been observed that Ang-2 affects the synthesis of cytokines by TEMs, e.g., it inhibits the release of proinflammatory TNF [5,13] and IL-12 cytokines which have strong antiangiogenic activity [13]. It has also been found that TEMs are responsible for suppressing T cell proliferation and in weakening T cell response, which are important in cancer control [10]. Some reports also indicate that Tie2 expression monocytes increase the formation of regulatory T cells (Treg). It is believed that this is directly related to IL-10 activity, anti-inflammatory cytokines released by this monocyte subpopulation [7,10,14].

In this study, we analyzed TEM in the peripheral blood of CLL patients. Currently, there are only a few reports on TEMs in CLL patients. It is indicated that the number of circulating TEMs is strongly associated with cytogenetic abnormalities, such as 17p deletion [14]. Numerous studies have confirmed the theory that monocytes/macrophages in the CLL microenvironment modulate leukemic B lymphocytes viability and survival [15,16,17]. Nevertheless, TEMs remain very poorly characterized in CLL patients. It is currently unclear to which extent TEM expression is related to other routinely measured CLL markers, and whether the finding can be of any clinical significance.

## 2. Materials and Methods

### 2.1. Patients and Samples

One-hundred-and-four peripheral blood (PB) samples were obtained from CLL patients (40 females and 64 males; median age, 65 years; range, 46–87 years) at the time of diagnosis and before any anticancer therapy, from June 2014 to September 2018, at the Department of Hematooncology and Bone Marrow Transplantation of the Medical University of Lublin (Lublin, Poland). CLL diagnosis was based on the recommendations of the International Workshop on Chronic Lymphocytic Leukemia (IWCLL) [18]. Clinical stages were determined according to the Rai classification system [19]: 43 patients were at Stage 0, 32 patients were at Stage I, 15 patients were at Stage II, 8 patients were at Stage III and 6 patients were at Stage IV. Stage 0 patients were classified into a low-risk group, stage I and II patients—into an intermediate-risk group, while stage III and IV patients—into a high-risk group. Table 1 presents the patients’ characteristics at the time of diagnosis. Twenty-one healthy volunteers donated their blood as control samples (HVs; 9 females and 12 males, aged from 36–80 years, median, 56 years). 

PB samples, collected in heparinized tubes, were used for mononuclear cells’ separation by density gradient centrifugation on Gradisol L (Cat No.: 9003.1, Aqua-Med, Łódź, Poland) for 25 min at 400× *g* at room temperature. Mononuclear cells at the interphase were removed, washed twice and resuspended in phosphate-buffered saline (PBS).

### 2.2. Flow Cytometry Analysis of TEM

In order to analyze TEM (defined as CD14^+^CD16^+^Tie2^+^ cells) by means of flow cytometry, peripheral blood mononuclear cells (PBMCs) were stained with a set of fluorescent-labeled monoclonal antibodies (MoAbs): mouse anti-human CD202b (Tie2/Tek) PE (Clone 33.1 (Ab33), Cat No.: 334206; BioLegend, San Diego, CA, USA), mouse anti-human CD14 V450 (Clone MφP9, Cat No.: 655114) and mouse anti-human CD16 FITC (Clone NKP15, Cat No.: 347523) (BD Biosciences, Franklin Lakes, NJ, USA). Furthermore, the appropriate amount of aqua fluorescent reactive dye (LIVE/DEAD Fixable Aqua Dead Cell Stain Kit; Cat No.: L34957; Thermo Fisher Scientific, Invitrogen, Waltham, MA, USA) was added to the antibody cocktail. After incubation in the dark for 30 min (room temperature; RT) and washing in 1% bovine serum albumin-enriched PBS, samples were directly analyzed using flow cytometry. A FACSCanto II instrument with FACSDiva Software (BD Biosciences, Franklin Lakes, NJ, USA) was used for data acquisition. In each case, both acquisition and analysis were performed on 100,000 events. The results were presented as the percentage of TEMs in peripheral blood CD14^+^CD16^+^ monocytes. For data analysis, Kaluza 2.1.1 (Beckman Coulter, Miami, FL, USA) was used.

### 2.3. Analysis of Intracellular IL-10 Expression in TEM

In 30 CLL patients, we examined an intracellular IL-10 expression by CD14^+^CD16^+^Tie2^+^ and CD14^+^CD16^+^Tie2^-^ cells. PBMCs were stained with MoAbs against cell-surface markers: anti-Tie2 (CD202b) PE, anti-CD14 V450 and anti-CD16 FITC. Following incubation for 20 min at RT, cells were fixed and permeabilized with with Cytofix/Cytoperm solution and Perm/Wash buffer (Cat No.: 554714, BD Biosciences, Franklin Lakes, NJ, USA), according to the manufacturer’s protocol. Then, the samples were intracellularly stained (20 min at RT) with mouse anti-human IL-10 PerCP-Cy5.5 (Clone JES3-9D7, Cat No.: 501418, BioLegend, San Diego, CA, USA). Finally, a FACSCanto II instrument was used for data acquisition and Kaluza 2.1.1 software was used for data analysis.

### 2.4. Tie2-Positive and Tie2-Negative Monocytes Sorting for RT-qPCR

In 30 CLL cases, CD14+ cells were selected into two populations due to the Tie2 expression. Tie2-positive and Tie2-negative monocytes were sorted using BD FACSAria II flow cytometer (BD Biosciences; Franklin Lakes, NJ, USA). PBMCs were labeled with anti-CD14 FITC (Clone MφP9, Cat No.: 347493; BD Biosciences, Franklin Lakes, NJ, USA) and anti-CD202b (Tie2/Tek) PE (Clone 33.1, Cat No.: 334206; BioLegend, San Diego, CA, USA). The purity of the sorted populations was confirmed by flow cytometry and reached more than 97%. Purified fractions were used for RNA isolation.

### 2.5. RT-qPCR for IL-10, and VEGF

Purified CD14+Tie2+ and CD14+Tie2- fractions were analyzed for IL-10 and VEGF mRNA expression by quantitative RT-PCR, according to the method described previously [20]. All molecular tests were based on TaqMan Gene Expression Assays (Thermo Fisher Scientific, Applied Biosystems, Inc., Waltham, MA, USA)—assay ID for IL-10 was Hs00961622_m1 and, for VEGF Hs00998133_m1, used with TaqMan Gene Expression Master Mix (Cat No.: 4369016). The β-actin gene expression served as an internal control (Human ACTB (Beta Actin) Endogenous Control, Cat No.: 4310881E, Thermo Fisher Scientific, Applied Biosystems, Inc., Waltham, MA, USA). Data are presented as 2^-ΔCq^. ΔCq is the difference between the cycle quantification value (Cq) of the target gene (Cqt) and the reference gene (Cqr) (ΔCq = Cqt−Cqr).

### 2.6. Co-Culture Conditions

We also evaluated whether leukemic cells had a direct effect on Tie2 expression, IL-10 or VEGF production in the monocyte fraction. For this, monocytes from six healthy donors were cultured in direct contact with CLL cells. Monocytes were isolated by plastic adherence. A total of 2 × 10^6^ PBMC were seeded in Nunc Cell-Culture Treated multi-well plates in RPMI 1640 medium (ThermoFisher Scientific Inc., Waltham, MA, USA) supplemented with 2% human albumin, and 100 IU of penicillin/streptomycin and allowed to adhere for 2 h at 37  °C and 5% CO_2_. After incubation, the nonadherent cells were removed by thorough washing with RPMI-1640.

Cryopreserved PBMC from six CLL patients were used for CD19+ B cells separation. B lymphocytes were isolated using the CD19 MicroBeads (Cat No.: 130-050-301; Miltenyi Biotec, Bergisch Gladbach, Germany). Magnetically labeled CD19+ cells were enriched by using the LS column. The purity of selected cells was over 99% as assessed by flow cytometry.

For coculture experiments with monocytes, purified CLL cells were plated on monocytes layers for 24 h. A ratio of 1 monocyte to 10 CLL cells was used. Monocytes were also cultured in the medium alone or were treated with 1 μg/mL lipopolisaharyde (LPS) from *Escherichia coli* (Sigma-Aldrich, Steinheim, Germany) for 24 h. LPS was used as a standard positive control. Monensin solution (GolgiStop; Cat No.: 554724, BD Biosciences, Franklin Lakes, NJ, USA) was added during the last 4 h of culture. To detach cultured monocytes from the cell culture plate for further experiments, they were recovered by intensive pipetting or carefully scraping, washed them and suspended in the PBS. The cultured monocytes were analyzed for IL-10 and VEGF mRNA expression by quantitative RT-PCR. The intracellular IL-10 expression in Tie2+ monocytes was also examined (by flow cytometry as described above).

### 2.7. Plasma Ang-2 Immunoassay

Plasma samples received from EDTA–anticoagulated blood sample was kept at –80°C. For a quantitative determination of human Ang-2 in plasma samples, a commercial enzyme-linked immunosorbent assay (ELISA) kits (Human Angiopoietin-2 Quantikine ELISA Kit, Cat No.: DANG20, R&D Systems, Inc., Minneapolis, MN, USA) was used, following manufacturer’s recommendations. The ELISA Reader Viktor^3^ (Perkin Elmer, Waltham, MA, USA) was used. A detection limit for Ang-2 was: 21.3 pg/mL.

### 2.8. Flow Cytometric Analysis of CD38 and ZAP-70 Expression in CLL Cells

In each case CLL cells were analyzed for CD38 antigen and ZAP-70 protein expression (as described previously [21]). Fresh PB samples were stained with the following set of monoclonal antibodies: FITC mouse anti-human CD19 (Clone SJ25C1, Cat No.: 340409), PE-Cy5 mouse anti-human CD5 (Clone UCHT2, Cat No.: 555354) and CD38 PE (Clone HIT2, Cat No.: 555460) or anti-ZAP-70 PE (Clone Clone 1E7.2, Cat No.: 344636) (BD Biosciences, Franklin Lakes, NJ, USA). Cells were considered positive for ZAP-70 with a cut-off point of ≥20%, and positive for CD38 expression with a cut-off point ≥30%.

### 2.9. Fluorescence In Situ Hybridization (FISH)

FISH method was used to analyze cytogenetic abnormalities in leukemic cells, as described previously [22]. The analysis was based on commercially available Vysis probes (Abbott Molecular Europe, Wiesbaden, Germany) were used: LSI TP53 SpectrumOrange/CEP 17 SpectrumGreen Probe and LSI ATM SpectrumOrange/CEP 11 SpectrumGreen Probe. For each probe at least 200 nuclei were analyzed, and the border value for positive results was 2.5%.

### 2.10. Analyses of IGHV Mutations

The isolation of genomic DNA was performed using the QIAamp DNA BloodMini Kit (Qiagen, Netherlands), following the manufacturer’s recommendation. V, D, and J rearranged genes were amplified basing on the BIOMED-2 references. The sequencing of PCR products was performed using an ABI PRISM BigDye Terminator v3.1 cycle sequencing kit (Applied Biosystems, Foster City, CA, USA) on an automatic ABI 3500 Genetic Analyzer (Applied Biosystems, Foster City, CA, USA The IMGT/V-QUEST analysis software was used to align each IGHV sequence to the closest matched germline gene (IMGT; http://www.imgt.org/, accessed date: 19 February 2019). Unmutated sequences were characterised by ≥98% homology, whereas mutated sequences were characterised by <98% homology. 

### 2.11. Statistical Analysis

For comparative analysis of the variables, the Kruskal–Wallis test with post hoc Dunn’s correction, Wilcoxon’s matched-pairs rank test or U Mann-Whitney test were used. The data are presented as the median and interquartile range (IQR: 25% percentile and 75% percentile). Spearman’s rank correlation coefficient was used for correlation analyses. In order to compare time to treatment (TTT) and overall survival (OS) between categorical groups, Kaplan–Meier curves were used. The determination of differences between groups was based on a log rank test. OS and TTT were determined from the date of diagnosis until the last follow-up/death and the date of initial treatment, respectively. Cox regression analysis was constructed to determine the hazard ratio (HR). All statistically significant variables (*p* ≤ 0.05), as found in the univariate analyses, were included in multivariate analysis based on a Cox proportional hazards model. Receiver operating characteristics (ROC) analysis was used to calculate the most significant cut-off value of TEM percentage that best distinguished ZAP-70-positive and ZAP-70-negative cases. Since ZAP-70 has been previously proven to be one of the most powerful prognostic factor, it was used in ROC curve analysis [21]. An area under the curve (AUC) was also estimated. Differences were considered statistically significant with a *p*-value ≤ 0.05. 

Statistical analysis was performed using Statistica 13 PL (StatSoft, Cracow, Poland) and graphs were managed using GraphPad Prism version 5 (GraphPad Software, San Diego, CA, USA).

## 3. Results

### 3.1. TEM Percentage in CLL Patients

Flow cytometry analysis was used for the identification of circulating TEM in the CLL patients (Figure 1a–f). Tie2 was mainly expressed in CD14^+^CD16^+^ cells (Figure 1f) rather than in CD14^+^CD16^−^ cells (Figure 1e). Consequently, CD14^+^CD16^+^Tie2^+^ monocytes were considered as TEMs (Figure 1f). 

We found significantly higher percentages of TEM in peripheral blood of CLL patients (median (IQR), 13.32 (6.25–23.05)%) than in normal subjects (median (IQR), 3.76 (2.46–6.48)%, *p* < 0.0001; Figure 2a). In the group of CLL patients, there was a weak positive correlation between the percentage of circulating TEM and WBC count (r = 0.298; *p* < 0.05) and lymphocyte count (r = 0.293; *p* < 0.05). Among CLL patients, no significant correlations between the frequency of TEM and platelet counts, hemoglobin concentration, serum LDH and β2-microglobulin levels or patients’ age were identified. A significant positive correlation was observed between the percentage of TEM and the number of monocytes (r = 0.301; *p* < 0.05).

In CLL patients TEM percentage was significantly higher in patients with the high-risk disease (Rai stages III–IV) (median (IQR), 34.58 (22.76–41.61)%) as compared to the low-risk disease (stage 0) (median (IQR), 8.04 (3.75–14.41)%) and intermediate-risk disease (stages I–II) (median (IQR), 14.82 (7.69–22.02)%) (Figure 2b).

### 3.2. TEM Percentage and CLL Adverse Prognostic Factors

The percentage of TEM was significantly lower in ZAP-70-negative patients (median (IQR), 8.39 (5.56–18.14)%) compared with ZAP-70-positive ones (median (IQR), 20.56 (10.07–34.40)%, *p* < 0.001; Figure 3a). Likewise, the percentage of TEM was lower in CD38-negative patients (median (IQR), 9.64 (5.91–18.90)%) than in CD38-positive (median (IQR), 18.90 (8.16–32.33)%), however the difference was not statistically significant (Figure 3b). Against, the group of patients carrying 11q22.3 and/or the 17p13.1 deletion showed higher percentages of TEM compared with patients without these adverse aberrations (median (IQR), 21.09 (7.54–45.16)% vs. 12.04 (6.09–21.33)%, *p* < 0.01; Figure 3c). Additionally, CLL patients with mutated IGHV genes (M-CLL) display a lower TEM percentage (median (IQR), 8.21 (5.84–17.6)%) than those with unmutated IGHV genes (U-CLL) (median (IQR), 20.56 (7.69–32.43)%) (*p* < 0.05) (Figure 3d). 

### 3.3. TEM Percentage and Clinical Outcome of CLL Patients

The median follow-up time was 45 months (range 1–71 months). During the follow-up period, 42 patients (40.4%) were treated. The percentage of Tie2-expressing monocytes measured at the time of diagnosis was significantly higher in patients requiring therapy (median (IQR), 20.38 (8.77–33.38)%) as compared with patients without treatment during the observation period (median (IQR), 9.37 (5.83–17.56)%, *p* < 0.001) (Figure 4a). Complete remission (CR) was achieved in 11 CLL patients (26.2%), partial remission (PR) in 18 patients (42.8%), and stable disease (SD) in 7 patients (16.7%) and disease progression (PD) were observed in 6 (14.3%) patients. In a patients with PD, the percentage of TEMs was significantly higher (median (IQR), 32.13 (28.36–38.36)%) than in those with CR (median (IQR), 8.07 (5.66–18.33)%, *p* < 0.05) (Figure 4b). There was no significant differences in TEM percentage between patients with CR and those with PR (median (IQR), 31.82 (19.69–37.29)%) or SD (median (IQR), 25.54 (4.08–37.50)%). Twenty one CLL-related deaths occurred during the observation period. Surviving patients showed a significantly lower TEM percentage than patients who died (median (IQR), 10.05 (5.82–20.88)% vs. 22.70 (14.15–34.76)%, respectively, *p* < 0.01; Figure 4c).

Ten patients requiring treatment were examined at three time-points: at the time of diagnosis, before the start of the treatment, and 6 or 12 months after chemotherapy. TEM percentage assessed over time is presented in Table 2. We noted that percentage of TEMs in individual patients changed over time and was significantly higher before the initiation of chemotherapy than at the time of diagnosis or after therapy (*p* < 0.01). However, differences between TEM percentage at the time of diagnosis and after treatment was not statistically significant (Table 2). 

### 3.4. Comparison of CLL TEM^high^ and TEM^low^ Patient Groups

To evaluate the clinical significance of TEM as CLL biomarkers, we compared different clinical and laboratory parameters in patients with either high or low TEM percentage. ROC curves were used to determine the most significant cut-off values of TEM and their prognostic influence. We determined that the optimum threshold for the percentage of TEM associated with ZAP-70 above 20% was 14.82% (AUC, 0.801; sensitivity, 79%; specificity, 76%; 95% confidence interval (CI), 0.709–0.894, *p* < 0.0001; Figure 5).

Using 14.82% as a cut-off value, we divided our patients into two groups: TEM^low^ (less than 14.82% CD14^+^CD16^+^Tie2^+^ monocytes; *n* = 57) and TEM^high^ (14.82% or more CD14^+^CD16^+^Tie2^+^ monocytes; *n* = 47). TEM^high^ and TEM^low^ patients’ characteristic at the time of CLL diagnosis is summarized in Table 1. As expected, TEM^low^ patients were more often (59.6%) at Rai stage 0. By contrast, in the TEM^high^ group, 19.1% of patients had a low-risk disease. Noteworthy, in the TEM^low^ group, the majority of patients (75.4%) were ZAP-70-negative or CD38-negative. We also noticed the notable prevalence of male patients (70.2%) in the TEM^high^ group (Table 1). The majority of the laboratory parameters did not differ significantly between groups. However, in the TEM^low^ group, we observed significantly lower lymphocyte, WBC and monocyte count than in the TEM^high^ group (*p* < 0.01). In TEM^high^ patients, the percentage of leukemic cells expressing ZAP-70 or CD38 was significantly higher than in TEM^low^ ones (*p* < 0.001) (Table 1). It is worth mentioning that, in the TEM^high^ group, 59.6% of patients required therapy. By contrast, in the TEM^low^ group, 75.4% of patients were without treatment during the observation period (Table 1). The median time to treatment (TTT) of patients with <14.82% and ≥14.82% TEM was 48 months and 29 months, respectively (*p* < 0.001). The median overall survival (OS) of patients with <14.82% and ≥14.82% TEM was 51 months and 40 months, respectively (*p* = 0.026).

### 3.5. The High TEM Percentage Is Associated with a Shorter Time to Treatment and Poor Overall Survival

TEM percentage above 14.82% showed a significant impact on time to treatment (hazard ratio (HR) 2.89; 95% CI 1.51–5.56; *p* < 0.001) (Figure 6a, Table 3). Variables included in the univariate analysis on TTT were age; genetic aberrations (del(17p13.1) and del(11q22.3)); IGHV mutational status; CD38 and ZAP-70 expression, and β2M level. Univariate Cox analysis selected ZAP-70 (HR 2.86; 95% CI 1.52–5.38; *p*  <  0.001), CD38 (HR 2.21; 95% CI 1.18–4.12; *p* = 0.012), *IGHV* mutational status (HR 0.36; 95% CI 0.15–0.86; *p* = 0.022) and B2M disruption (HR 6.02; 95% CI 3.17–11.46; *p* < 0.0001) as risk factors for shorter TTT, and these four parameters were included in the multivariate analysis in the next step (Table 3). The multivariate analysis by Cox proportional hazard regression confirmed TEM percentage (HR 2.57; 95% CI 1.34–4.97; *p* = 0.004) along with B2M (HR 5.66; 95% CI 2.95–10.86; *p*  < 0.0001) were independently correlated with shorter TTT in CLL patients (Table 3).

We also found a significant association between the percentage of TEM above 14.82% and the overall survival (OS) (HR 2.60; 95% CI 1.05–6.45; *p* = 0.026) (Figure 6b, Table 4). Univariate Cox analysis selected ZAP-70 (HR 2.16 (0.91–5.14); *p* = 0.038), CD38 (HR 3.11; 95% CI 1.29–7.53; *p* =  0.012) and B2M disruption (HR 18.20; 95% CI 5.34–62.02; *p* <  0.0001) as risk factors for shorter OS, and these three parameters were used in the multivariate analysis. However, in multivariate analysis, a shorter OS was not significantly associated with TEM percentage (HR 0.62; 95% CI 0.23–1.74; *p* =  0.372).

Our study also aimed to examine the significance of TEM percentage in predicting treatment outcome. As described above, 42 patients were evaluated for response to treatment. Eleven (26.2%) patients obtained a complete response (CR) and 18 (42.8%) a partial response (PR). The overall response rate (ORR), defined as CR plus PR, was 69%. However, no statistically significant differences were noticed in terms of ORR between the TEM^high^ and TEM^low^ group (62% vs. 84%, respectively). Eighteen patients received a second line of treatment at a median time of 38 months (range 2–48 months). Time to re-treatment (TTR), calculated from the first day of treatment until initiation of a new treatment, was shorter among the TEM^high^ cases (median 37, range 2–47 months) compared with TEM^low^ (median 42, range 12–48 months). However, the difference was not statistically significant (*p* > 0.05).

### 3.6. IL-10 and VEGF Are Overexpressed by CLL TEM

Tie2+ (TEM) and Tie2- monocytes were analyzed for intracellular IL-10 expression by flow cytometry. Representative dot plots from CLL patient and the gating strategy is presented in Figure 7a–c. The analysis was performed on freshly isolated PBMC (without culture or TEM stimulation). We found TEM expressed significantly more IL-10 protein than Tie2- monocytes (median (IQR), 2.38 (1.18–5.68)% vs. 0.48 (0.24–2.52)%, *p* < 0.01; Figure 7d). 

RT-qPCR was used to quantify the levels of mRNA expression for IL-10 and VEGF in purified Tie2+ and Tie2- monocytes. RT-qPCR confirmed the presence of IL-10 transcripts in both fractions. Consistent with flow cytometry, IL-10 mRNA level were significantly higher in the Tie2^+^ as compared to the Tie2^-^ fraction (median (IQR), 0.52 (0.04–1.49)% vs. 0.25 (0.005–1.06)%, *p* < 0.05; Figure 8a). There was a positive correlation between the intracellular IL-10 expression and IL-10 mRNA expression in Tie2-positive (r = 0.781; *p* < 0.01) and Tie2-negative (r = 0.538; *p* < 0.01) monocytes. Moreover, TEMs express higher levels of VEGF mRNA than Tie2^−^ monocytes (median (IQR), 0.45 (0.16–1.52)% vs. 0.8 (0.05–1.07)%, *p* < 0.05; Figure 8b).

The number and activity of TEMs are probably directly dependent on Ang-2 [7,10]. Therefore, we decided to assess its concentration in the plasma of CLL patients. Ang-2 concentration in peripheral blood plasma of CLL patients ranged from 1034.21 pg/mL to 7788.41 pg/mL (median (IQR): 2476 (2058–3110) pg/mL). We found a weak positive correlation between Ang-2 plasma and the percentage of TEM in CLL patients (r = 0.277; *p* < 0.05). There was a moderate correlation between Ang-2 concentration and the percentage of TEM with intracellular IL-10 expression (r = 0.451; *p* < 0.01). Moreover, a positive correlation was found between VEGF mRNA expression and plasma Ang-2 concentration (r = 0.441; *p* < 0.05).

### 3.7. The Direct Effect of CLL Cells on IL-10 or VEGF Expression in Healthy Monocytes

We also evaluated whether leukemic cells had a direct effect on IL-10 or VEGF production in the monocyte fraction. For this, monocytes from healthy donors were cultured in direct contact with CLL cells for 24 h. IL-10 and VEGF expression in monocytes cultured alone or monocytes cultured in direct contact with CLL cells is presented in Figure 9a–d. The analysis was also performed on LPS-stimulated monocytes. In the absence of CLL cells or LPS, only little intracellular IL-10 production in Tie2+ monocytes was detectable. Further analysis of mRNA expression for IL-10 or VEGF showed the same results. However, only the difference between LPS-stimulated monocytes and basal expression of IL-10 or VEGF was statistically significant (*p* < 0.05; Figure 9d). 

No statistically significant difference was detected in the TEM percentage between co-culture system (median (IQR), 2.86 (0.83–4.69)%. *p* > 0.05) and healthy monocytes cultured alone (median (IQR), 2.79 (0.99–5.13)%).

## 4. Discussion

The role of the immune system in modulating the biology and development of hematologic malignancies is now well recognized. In some cases, cancer cells share a development niche with normal immune cells. Mutual interactions between normal immune cells and tumor cells may affect tumor growth and survival [23,24,25]. It has been suggested that circulating blood monocytes, components of the tumor microenvironment may promote CLL cell survival [17]. Among monocytes, those with Tie2 angiopoietin receptor expression have tumor promoting properties [14].

In the present study, the frequency of CD14^+^CD16^+^Tie2^+^ monocytes (TEM) in peripheral blood of CLL patients has been investigated. Our data confirm previous findings [5,14] that human monocytes can be divided into two subsets based on their expression of CD14 and CD16 antigens. Our flow cytometry analysis showed that CD16-negative monocytes express Tie2 to a lower degree than CD16-positive monocytes. The data obtained in the current study are concordant with those reported by Murdoch et al. [5] who demonstrated that Tie2 is expressed at higher levels by monocytes with a CD14^high^/CD16^high^ than a CD14^high^/CD16^low^ phenotype. Tie2 expression was almost undetected on CD16-negative monocytes [5]. Likewise, consistent with our study, Xue et al. [26] revealed that Tie2 was mainly expressed in CD14^+^CD16^+^ cells rather than in CD14^+^CD16^−^ cells. 

Measuring TEM by flow cytometry, we showed higher cell percentages in CLL patients than in healthy controls. The results of the current study are consistent with those of Maffei et al. [14], who reported higher frequencies of TEM in peripheral blood of CLL patients. TEM and their role in a tumor microenvironment have been examined by several groups in the different cancer setting. However, the research has mainly focused on solid tumors [4,6,27,28]. The presence of TEM has only been described in several hematological tumors [14,29], and their importance is increasing. It is worth noting that a significant proportion of acute myeloid leukemia (AML) patients have circulating TEM [29].

In our study, TEM percentage correlated significantly with the Rai stage of the disease. By contrast, another research group detected no significant differences between the disease stages and TEM expression [14]. Additionally, no association between TEM number, white blood cell count, and CD38 or ZAP-70 expression has been found in their study [14]. It should be noted, however, that the researchers assessed fewer patients (*n* = 26) as compared to the number of patients tested in our study (*n* = 104). It must be emphasized that a higher percentage of TEM was observed in patients carrying unfavorable cytogenetic changes: deletion 17p13.1 and/or deletion 11q22.3 compared with the percentage of TEMs in patients without these adverse aberrations. These results are in agreement with the results of Maffei et al. [14] who showed higher TEM’ expression in patients with 17p deletion. Close association with unfavorable prognostic markers (i.e., ZAP-70, CD38, IGHV mutational status, 11q and 17p deletion), observed in our study, suggests a potential role of TEM as a prognostic factor. TEM percentage measured at the time of diagnosis was significantly higher in patients requiring therapy in comparison with patients without treatment during the observation period. Moreover, the percentage of TEMs was significantly higher in patients not responding to the first-line therapy as compared to responding patients, suggesting its potential predictive value. We observed that the percentage of TEMs changed over time and was significantly higher before the initiation of chemotherapy comparing to the values at diagnosis. However, at 6 or 12 months after therapy, the TEM percentage did not increase. Rather, there was a decline after treatment. The limitation of our study is the small number of patients. Although the results among CLL patients are promising, larger trials with longer follow-up are needed. Nevertheless, it seems that increased TEM expansion occurs with disease progression.

We established an association between a high TEM number and a shorter time to treatment. CLL patients with a proportion of 14.82% TEMs (according to the ROC analysis) or more had a shorter TTT than the group with lower percentages of Tie2-positive monocytes. The high percentage of TEMs in the peripheral blood was also associated with poor overall survival. In the multivariate analysis baseline TEM percentage held its independent prognostic factor for time to treatment. This is an important newness of the current study. In our study, the overall response rate or time to re-treatment did not differ significantly between the TEM^low^ and TEM^high^ group. However, this preliminary study enhances our understanding of TEM as a part of the CLL microenvironment that can contribute to CLL progression. TEM have been associated with a worse prognosis in many cancer types. In renal cell carcinoma (RCC), TEM correlate with tumor grade, lymph node and distant metastases [28]. Matsubara et al. [30] found that TEM are significantly increased in the peripheral blood and liver of hepatocellular carcinoma (HCC) and can be used as a diagnostic marker of HCC, potentially reflecting angiogenesis in the liver. TEM frequency showed a correlation with NSCLC (non-small cell lung cancer) recurrence. The high percentage of TEMs in the peripheral blood was associated with a lower survival rate [26]. Likewise, the interactions between CLL cells and their microenvironment in bone marrow, lymph nodes and peripheral blood are believed to have a significant impact on the pathophysiology of the disease [14,31].

We found Tie2+ monocytes expressed significantly more VEGF mRNA than Tie2- monocytes. This is not altogether surprising considering the stimulation of angiogenesis is probably the key role of TEM [3,5,9]. The importance of the process has been described not only in the development of solid tumors but also in the pathomechanism of lymphoproliferative tumors, including CLL [32,33,34]. CLL patients demonstrate increased vascularization density of lymph nodes [32] and bone marrow [33,35]. Increased angiogenesis in bone marrow has been associated with advanced disease stage and aggressiveness [33,35]. It is known that TEMs express increased levels of several proangiogenic factors, such as vascular endothelial growth factor (VEGF) [7] and basic fibroblast growth factor (bFGF) [3,6]. Both bFGF and VEGF are the most studied angiogenic factors in cancer research. Increased bFGF and VEGF levels in CLL patients correlate with poorer prognosis [36,37]. In addition, high levels of the pro-angiogenic factor Ang-2 have been found in the plasma of CLL patients and correlated with adverse prognosis [36,37]. It has been shown that in human peripheral blood TEM respond to Ang-2 using the Tie2 receptor [4,5]. Moreover, CLL cells can secrete Ang-2 [36]. In accordance with the research of Maffei et al. [14], we found a positive correlation between the percentage of TEMs and plasma Ang-2 concentration. High Ang-2 levels have been associated with a shorter time to treatment and rapid disease progression [14,38,39]. Coffelt et al. [10] suggested that Ang-2 also stimulates the expression of immunosuppressive factors by TEM. They found TEM expressed significantly more IL-10 protein than TAM (tumor-associated macrophages) [10]. It should be noted that in our research Ang-2 concentration positively correlated with the percentage of TEM with intracellular IL-10 expression. In our study, the immunosuppressive cytokine IL-10 is upregulated by TEM. Moreover, a tendency towards the induction of IL-10 and VEGF expression in Tie2 monocytes from healthy donors by CLL cells was observed. In the absence of CLL cells or LPS, only little IL-10 or VEGF production was detectable. Our data address the effects of CLL cells on allogeneic normal monocytes in vitro and may not show the in vivo situation. However, it appears that the TEM function may depend on CLL cells. As mentioned, CLL cells can secrete Ang-2 [36]. Furthermore, Coffelt et al. [10] found that Ang-2 stimulates TEM to express high levels of IL-10 to suppress antitumor immunity and to expand the T regulatory (Treg) population (CD4^+^CD25^+^FoxP3^+^). These authors suggest that TEM promote tumor progression not only by the stimulation of angiogenesis but also by the inhibition of antitumor immunity and the induction of tolerance via expansion of Treg lymphocytes [10]. Likewise, Huang et al. [40] reported that TEM are responsible for suppressing T cell proliferation and expand Treg by Ang-2 induced IL-10, which might promote tumor progression [35]. CLL demonstrates Treg accumulation and the percentage of these lymphocytes is often associated with clinical progression and worse prognosis [41].

Our results confirm that a relevant part of CLL pathogenesis is its microenvironment where different components impact the disease course and justify its clinical heterogeneity. Different cell types may provide a solid base for identifying novel targets of treatment. Tie2-expressing monocytes appear also to be such a target. In CLL mouse models, Galletti et al. [42] found that monocyte/macrophage pool depletion impairs CLL engraftment, inhibits disease progression, and favorably impacts mouse survival.

## 5. Conclusions

In conclusion, our data indicate that circulating TEMs are compartment of the tumor-microenvironment that could take part in disease progression. TEMs can be an important complement to other prognostic indicators.

## Figures and Tables

**Figure 1 cancers-13-02817-f001:**
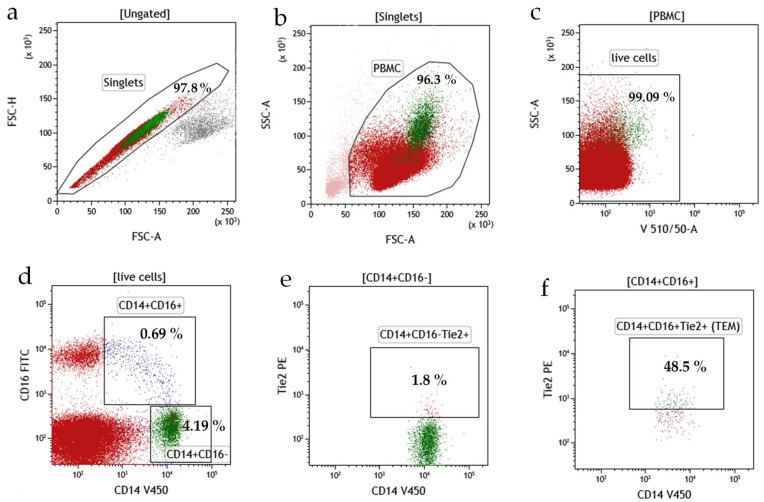
Representative dot plots from a CLL case illustrating the analytic method for the identification of TEMs (CD14+CD16+Tie2+). (**a**) FSC-A vs. FSC-H dot plot: doublets’ discrimination. (**b**) After gating a singlet PBMC were selected based on their SSC/FSC properties. (**c**) Gated events were analyzed for Aqua fluorescent reactive dye staining (LIVE/DEAD Fixable Dead Cell Stain Kit). Discrimination of live and dead cells. (**d**) Gated live cells were analyzed for CD14 V450 and CD16 FITC staining (dot plot: CD14 V450 vs. CD16 FITC). CD16-positive monocytes were gated (CD14+CD16+). CD16-negative monocytes were also analyzed (CD14+CD16−). Selected CD14+CD16+ and CD14+CD16− monocytes were analyzed for Tie2 PE staining. (**e,f**) Final dot plots (CD14 V450 PE vs. Tie2) indicate CD14+CD16− (**e**) and CD14+CD16+ (**f**) monocytes positive for Tie2 expression. TEM were defined as CD14+CD16+Tie2+ cells (**f**). The gate for Tie2+ cells was set based on the FMO control. FMO control contained all fluorochromes in a panel except for Tie2 PE. The FMO control identified any spread of fluorochrome in an unlabeled channel and places the gates in the correct place. Each dot plot shows the input gate in the title. The percentage for each gate was shown. Data were analyzed using Kaluza 2.1.1 software. TEM, Tie-2 expressing monocytes; FSC, forward scatter; SCC side scatter; PBMC, peripheral blood mononuclear cells; FMO, Fluorescence Minus One.

**Figure 2 cancers-13-02817-f002:**
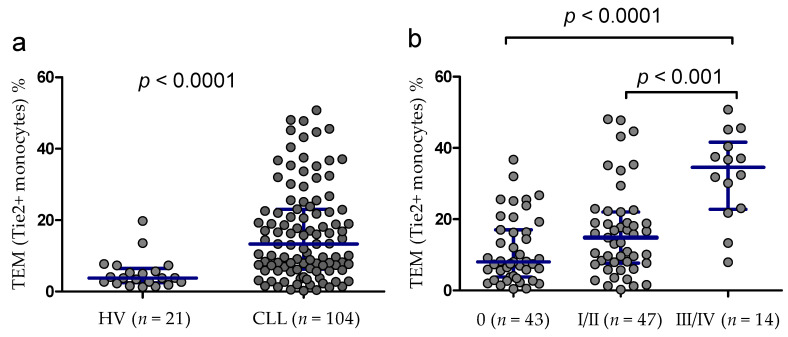
Percentages of Tie2+ monocytes (TEM) in CLL patients and healthy volunteers (HV) (**a**). The percentage of TEMs in peripheral blood of CLL patients at various disease stages (three risk groups) (**b**). Scatter plots present raw data. The central line shows the median. “Whiskers” represent from the first quartile to the third quartile (IQR, interquartile range).

**Figure 3 cancers-13-02817-f003:**
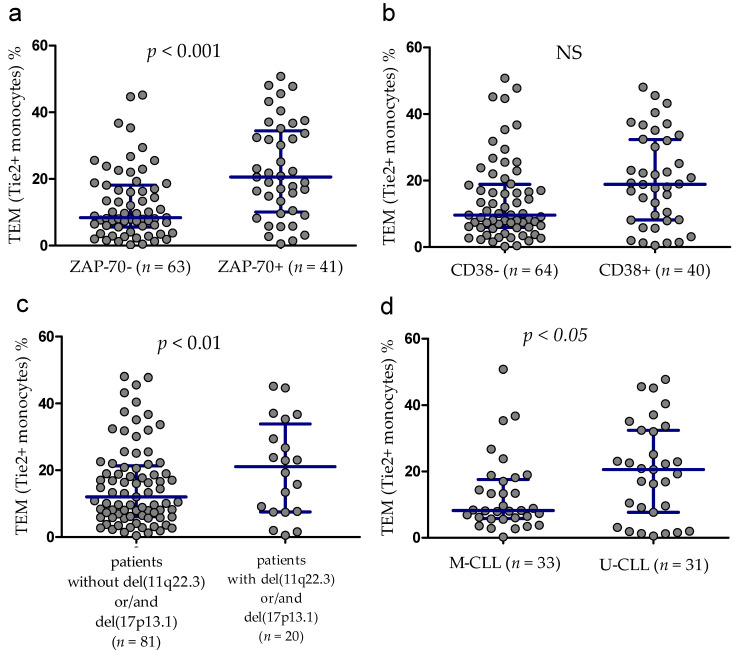
Tie2+ monocytes (TEM) percentage and adverse prognostic factors. (**a**) ZAP-70+ and ZAP-70− patients and (**b**) CD38+ and CD38− ones. (**c**) TEM percentage in CLL patients carrying the unfavorable cytogenetic changes and patients without adverse aberrations. (**d**) Tie2+ monocytes in patients harboring different status of IGHV mutation. Scatter plots present raw data. The central line shows the median. “Whiskers” represent IQR. IGHV mutational status was accessible for 64 participants of this study. U-CLL, IGHV-unmutated; M-CLL, IGHV-mutated, IQR, interquartile range.

**Figure 4 cancers-13-02817-f004:**
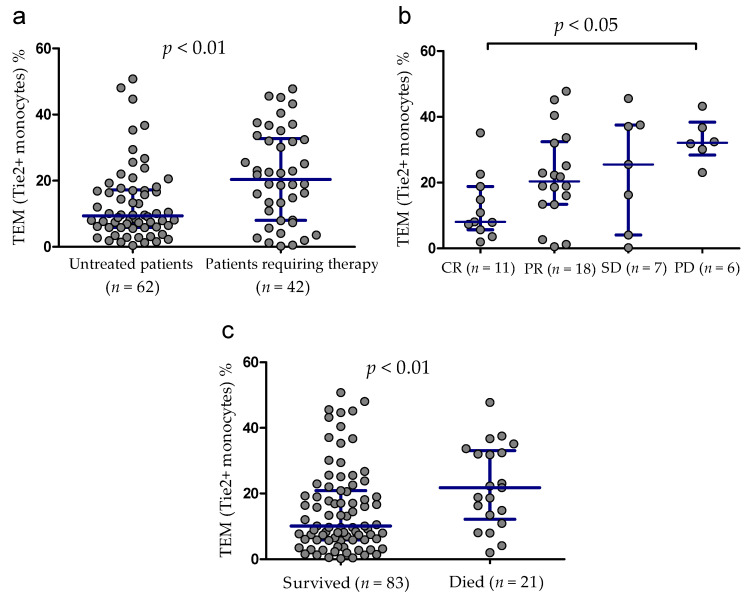
Tie2+ monocytes (TEM) percentage in patients requiring therapy as compared to patients without treatment during the observation period (**a**). The percentage of TEMs in patients responding to treatment and those with progressive disease (**b**). TEM percentage in CLL patients who survived and in the group that died during the observation period (**c**). Scatter plots present raw data. The central line shows the median. “Whiskers” represent from the first quartile to the third quartile (IQR, interquartile range). CR, complete response; PR, partial response; SD, stable disease; PD, progressive disease.

**Figure 5 cancers-13-02817-f005:**
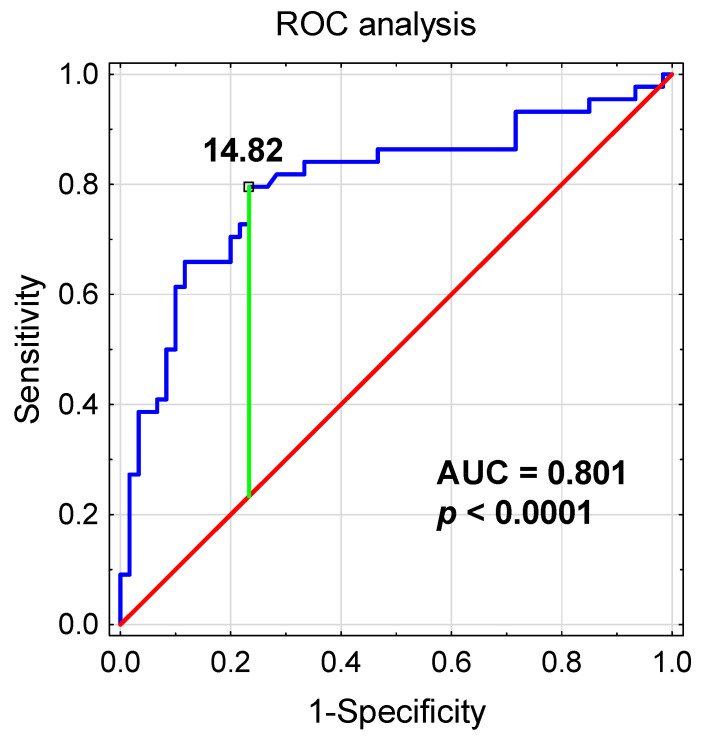
ROC and AUC were used to evaluate the sensitivity and specificity and to calculate the most significant cut-off value of TEM percentage that best distinguished ZAP-70-positive and ZAP-70-negative cases. ROC, receiver operating characteristic; AUC, area under the curve; TEM, Tie-2 expressing monocytes.

**Figure 6 cancers-13-02817-f006:**
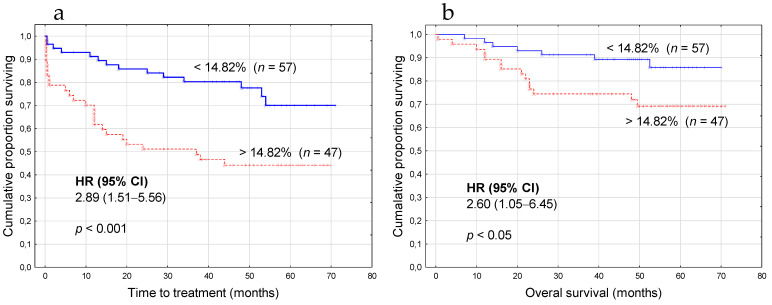
Kaplan–Meier curves based on a cut-off value of 14.82% for a TEM percentage, comparing TTT (time to treatment) among patients with CLL (**a**). Kaplan–Meier comparing OS (overall survival) among CLL patients (**b**). HR, hazard ratio; CI, confidence interval.

**Figure 7 cancers-13-02817-f007:**
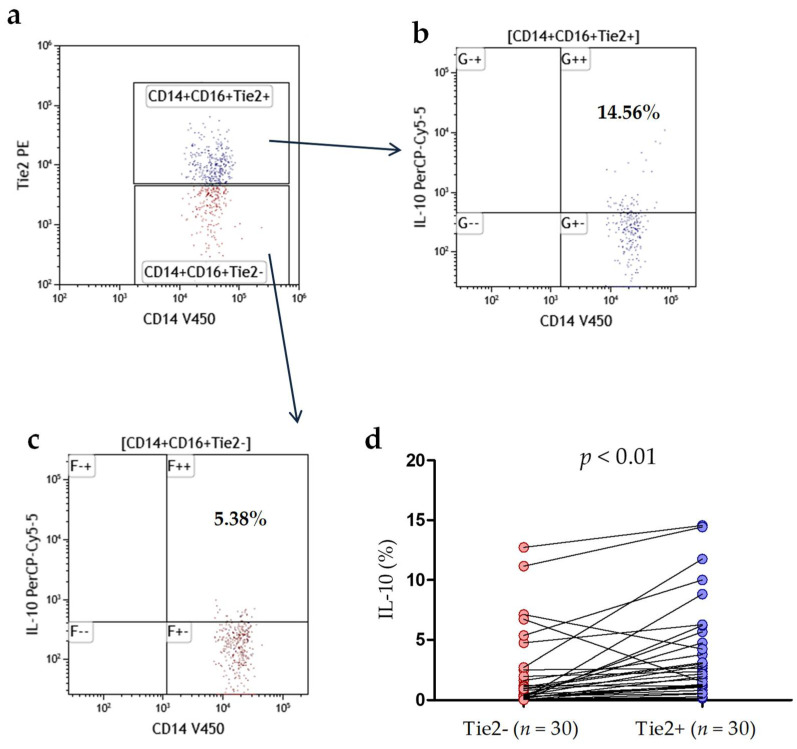
Intracellular IL-10 expression in CD14+CD16+Tie2+ (TEM) and CD14+CD16+Tie2- monocytes. (**a**) The dot plot (CD14 V450 vs. Tie2 PE) indicate Tie2-positive cells (TEMs; blue). Additionally, the population CD14+CD16+ without Tie2 expression was gated (CD14+CD16+Tie2-; red). CD14+CD16+ monocytes were selected as was shown in Figure 1a–f. Then the Tie2-positive (TEM) (**b**) and Tie2-negative (**c**) cells were analyzed for intracellular IL-10 expression. FMO (Fluorescence Minus One) control was used to set proper gates. Data were analyzed using Kaluza 2.1.1 software. (**d**) Percentages of Tie2-positive (TEMs) and Tie2-negative cells with intracellular IL-10 expression in 30 CLL patients. TEM, Tie-2 expressing monocytes; SCC side scatter.

**Figure 8 cancers-13-02817-f008:**
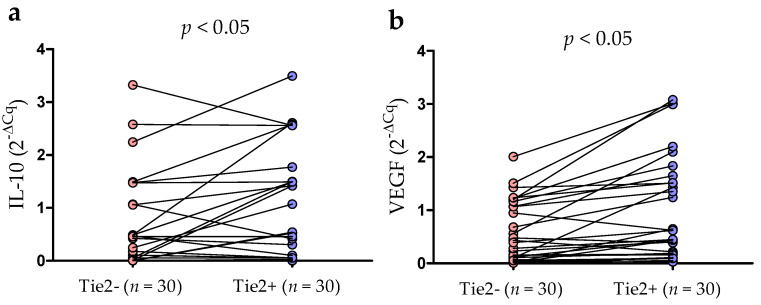
Quantitative expression of IL-10 (**a**) and VEGF (**b**) mRNA. RT-qPCR was performed on RNA samples isolated from Tie2+ (TEM) and Tie2- monocytes obtained from CLL patients (*n* = 30). Data are presented as 2^-ΔCq^. ΔCq is the difference between the Cq of the target gene (Cqt) and the reference gene (Cqr) (ΔCq = Cqt−Cqr). TEM, Tie-2 expressing monocytes; Cq, cycle quantification value.

**Figure 9 cancers-13-02817-f009:**
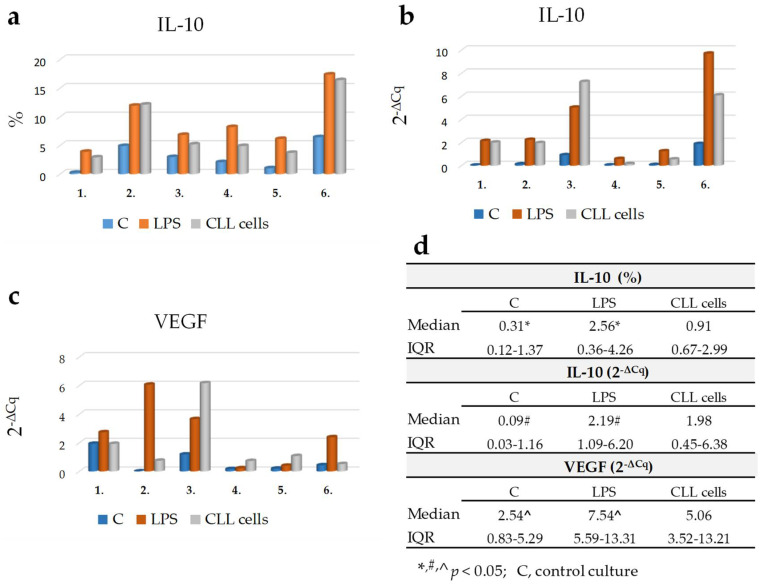
The effect of CLL cells on IL-10 or VEGF expression in monocytes from six healthy donors. (**a**) Percentage of Tie2+ monocytes with intracellular IL-10 expression (as assessed by flow cytometry). (**b**) Quantitative expression of IL-10 mRNA and (**c**) VEGF mRNA (data are presented as 2^-ΔCq^). (**d**) Statistical analysis. The data are presented as the median and interquartile range (IQR: 25% percentile and 75% percentile). Monocytes from healthy donors were cultured alone (basal expression; control culture (C) without stimulation; blue column), LPS—stimulated (positive control; red column) or cultured in direct contact with CLL cells (grey column). Each number under the graphs represents a set of related data for an individual healthy donor.

**Table 1 cancers-13-02817-t001:** Baseline characteristics at the time of diagnosis for CLL patients.

Variable	All patients	TEM^low^ < 14.82%	TEM^high^ > 14.82%
No. of patients	104	57 (54.8)	47 (49.2)
Sex			
Female *	40 (48.5)	26 (45.6)	14 (29.8)
Male *	64 (61.5)	31 (54.4)	33 (70.2)
Risk groups			
Low-risk (Stage 0) *	43 (41.3)	34 (59.6)	9 (19.1)
Intermediate-risk (Stages I–II) *	47 (45.2)	21 (36.8)	26 (55.3)
High-risk (Stages III–IV) *	14 (13.5)	2 (3.6)	12 (25.6)
ZAP-70 (cut-off 20%)			
Positive *	41 (39.4)	14 (24.6)	27 (57.4)
Negative *	63 (60.6)	43 (75.4)	20 (42.6)
CD38 (cut-off 30%)			
Positive *	40 (38.5)	14 (24.6)	26 (55.3)
Negative *	64 (61.5)	43 (75.4)	21 (44.7)
Cytogenetic abnormalities			
del(17p13.1) *	8 (7.7)	4 (7.0)	4 (8.5)
del(11q22.3) *	10 (9.6)	3 (5.3)	7 (14.9)
del(17p13.1) and del(11q22.3) *	2 (1.9)	0 (0.0)	2 (4.3)
Without del(17p13.1) and del(11q22.3) *	81 (77.9)	49 (86.0)	30 (63.8)
Not evaluated	3 (2.9)	1 (1.7)	2 (4.3)
IGHV mutation status			
Unmutated *	31 (29.8)	12 (21.1)	19 (40.4)
Mutated *	33 (31.7)	24 (42.1)	9 (19.2)
Not evaluated *	40 (38.5)	21 (36.8)	19 (40.4)
Patients requiring therapy *	42 (40.4)	14 (24.6)	28 (59.6)
Untreated patients *	62 (59.6)	43 (75.4)	19 (40.4)
No. of Deaths *	21 (20.2)	7 (12.3)	15 (31.9)
Age at diagnosis (years) ^$^	65 (46–87)	65 (46-87)	65 (46-84)
WBC count (G/L) ^#^	26.2 (18.23–48.53)	23.61 (17.31–35.46)	42.2 (21.39–76.49)
Lymphocyte count (G/L) ^#^	20.7 (11.57–44.22)	18.26 (10.99–26.25)	35.7 (12.90–56.61)
LDH (IU/L) ^#^	374 (331–410)	363 (323–392)	390 (331–426)
Hemoglobin (g/dL) ^#^	13.9 (12.78–14.80)	13.9 (12.78–14.80)	14.0 (12.70–14.80)
Platelets (G/L) ^#^	186 (150–223)	198 (167–236)	165 (134–212)
β2M (mg/dL) ^#^	2.4 (1.90–3.31)	2.25 (1.85–2.95)	2.9 (1.95–4.51)
CD19+/CD5+/ZAP-70+ cells (%) ^#^	19.2 (8.16–26.60)	12.2 (7.34–19.79)	21.1 (13.53–33.81)
CD19+/CD5+/CD38+ cells (%) ^#^	9.7 (2.03–45.60)	5.19 (1.50–28.75)	29.8 (4.58–50.07)

* Number (percentages), ^$^ median (range), ^#^ median (IQR); IGHV, immunoglobulin heavy chain variable gene; WBC, white blood cell; LDH, lactate dehydrogenase; and β2M, β2-microglobulin; IQR, interquartile range.

**Table 2 cancers-13-02817-t002:** TEM percentage in CLL patients over time. Each row in a table represents a set of related data for an individual patient.

	TEM (%)
Patient No.	At the Time of Diagnosis	Before Treatment	After Treatment *
1.	30.11	35.18	14.52
2.	2.68	3.48	2.81
3.	37.07	43.38	35.78
4.	25.54	28.04	NE
5.	3.57	3.64	2.81
6.	7.94	9.04	3.89
7.	4.08	5.66	5.48
8.	7.32	8.04	6.69
9.	16.03	17.38	10.14
10.	23.09	27.38	8.97
Median	11.99	13.21	6.69
IQR	3.95–26.68	5.15–29.83	3.35–12.33
	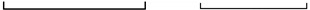 *p* < 0.01 *p* < 0.01

* Blood samples were isolated from CLL patients at 6 (patient no. 1–3) or 12 (patient no. 5–10) months after treatment. IQR, interquartile range; TEM, Tie2-expressing monocytes; NE, not evaluated.

**Table 3 cancers-13-02817-t003:** Univariate and multivariate analysis for time to treatment.

	Univariate			Multivariate	
Variable	Median TTT (Months)	HR (95% CI)	*p*	HR (95% CI)	*p*
Age					
≥65 years	44	1.50 (0.48–2.88)	0.215		
<65 years	47				
ZAP-70					
≥20%	29	2.86 (1.52–5.38)	**<0.001**	1.51 (0.49–4.66)	**0.048**
<20%	50				
CD38					
≥30%	38	2.21 (1.18–4.12)	**0.012**	0.98 (0.35–2.76)	0.231
<30%	48				
β2M					
≥3.5 mg/dL	10	6.02 (3.17–11.46)	**<0.0001**	5.66 (2.95–10.86)	**<0.0001**
<3.5 mg/dL	50				
del(17p13.1) or del(11q22.3)					
Positive	34	1.49 (0.65–3.38)	0.340		
Negative	47				
*IGHV* mutation status ^#^					
Unmutated	29	0.36 (0.15–0.86)	**0.022**	0.64 (0.23–1.82)	0.407
Mutated	47				
TEM					
≥14.82%	29	2.89 (1.51–5.56)	**<0.001**	2.57 (1.34–4.97)	**0.004**
<14.82%	48				

TTT, time to treatment; β2M, β2 microglobulin; *IGHV*, immunoglobulin heavy chain variable gene; TEM, Tie2-expressing monocytes; HR, hazard ratio; 95% CI: 95% confidence interval. ^#^ One-hundred-and-ten patients were selected for the analysis. However *IGHV* mutational status was accessible for 64 participants of this study. The *p*-values ≤  0.05 are shown in bold. Only variables with *p* < 0.05 in the univariate analysis were added to the multivariate analysis.

**Table 4 cancers-13-02817-t004:** Univariate and multivariate analysis for overall survival.

	Univariate			Multivariate	
Variable	Median OS (Months)	HR (95% CI)	*p*	HR (95% CI)	*p*
Age					
≥65 years	50	1.46 (0.34–6.13)	0.603		
<65 years	51				
ZAP-70					
≥20%	48	2.16 (0.91–5.14)	**0.038**	0.95 (0.34–2.71)	0.362
<20%	52				
CD38					
≥30%	46	3.11 (1.29–7.53)	**0.012**	1.21 (0.42–3.53)	0.724
<30%	54				
β2M					
≥3.5 mg/dL	38	18.20 (5.34–62.02)	**<0.0001**	15.83 (4.45–56.43)	**<0.0001**
<3.5 mg/dL	54				
del(17p13.1) or del(11q22.3)					
Positive	40	1.64 (0.55–4.87)	0.369		
Negative	52				
*IGHV* mutation status ^#^					
Unmutated	41	0.39 (0.12–1.32)	0.132		
Mutated	51				
TEM					
≥14.82%	40	2.60 (1.05–6.45)	**0.026**	0.62 (0.23–1.74)	0.372
<14.82%	51				

OS, overall survival; β2M, β2 microglobulin; *IGHV*, immunoglobulin heavy chain variable gene; TEM, Tie2-expressing monocytes; HR, hazard ratio; 95% CI: 95% confidence interval. ^#^ One-hundred-and-ten patients were selected for the analysis. However *IGHV* mutational status was accessible for 64 participants of this study. All statistically significant variables (*p* ≤ 0.05), as found in the univariate analyses, were included in multivariate analysis based on a Cox proportional hazards model. The *p*-values ≤  0.05 are shown in bold.

## Data Availability

The data presented in this study are available within the article. Other data that support the findings of this study are available upon request from the corresponding authors.

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
