# Peer review of "Prognostic Value of Tie2-Expressing Monocytes in Chronic Lymphocytic Leukemia Patients"

_cancers, 2021, doi:10.3390/cancers13112817_

Round 1

Reviewer 1 Report

In this manuscript Wos et al. investigated the prognostic value of Tie-2 expressing monocytes in CLL patients. They demonstrate that an increased proportion of Tie-2 expressing monocytes correlates with several well established poor prognostic markers of CLL and is predictive of shorter time to first treatment and poorer overall survival. The study is novel being the first to describe this association but is essentially descriptive and correlative. I have the following specific comments

Major comments

  • Was TEM high/low status predictive of response to therapy?
  • Does % TEM cells increase with disease progression in the same patient - ie sequential samples?
  • Is % TEM altered by treatment?
  • It would be useful to demonstrate a direct effect of CLL cells on Tie2 expression/IL-10/VEGF production in the monocyte fraction - e.g. Does co-culture (direct or transwell) of CLL cells with healthy (allogeneic) monocytes affect the expression of these molecules? Can this be replicated by purified Ang-2?
  • Do CLL TEM have lower levels of TNF and IL-12?

Minor comments

  • Methods: How were the CD14+Tie2+/- fractions purified?
  • Results: What is the denominator for Tie2+ monocytes? PBMCs? CD19- cells? CD14+ cells?
  • Minor English language editing would improve the feel and flow of the manuscript

Author Response

Dear Reviewer 1, thank you very much for your invaluable suggestions. We did our best to address them all properly during that short time we were given for revisions. We hope that you would agree that after implementing your suggestions and those of Reviewer 3, the overall quality of our paper increased significantly. Please find the more detailed answers to your suggestions and questions below.

Major comments

Point 1. Was TEM high/low status predictive of response to therapy?

Response 1: Decreased TEM percentage was detected in the group of patients with a confirmed response (CR) in comparison with patients in the group without clinical response (PD) ("Results" section; point 3.3. "TEMs percentage and clinical outcome of CLL patients"). However, our limitation is the small number of patients who show a robust response to the treatment. Although the results among CLL patients are promising, larger trials with longer follow-up are needed. However, it seems that as the disease progresses, there is an increase in the TEM percentage. TEM can be considered a dynamic time-dependent marker ("Results" section; point 3.3. "TEMs percentage and clinical outcome of CLL patients"; line 278-281). The Reviewer's suggestion is a very good idea for our future research in a larger cohort of patients.

Point 2. Does % TEM cells increase with disease progression in the same patient - ie sequential samples?

Response 2: Ten CLL patients requiring treatment were studied at two time-points: at the time of diagnosis and before the start of the treatment. The TEM percentage was significantly higher before the initiation of chemotherapy comparing to the values at diagnosis. It seems that increased TEM expansion occurs with disease progression. We added this issue into a revised version of our manuscript (“Results” section; part: 3.3. “TEMs percentage and clinical outcome of CLL patients”, line 278-281). TEM percentage assessed over time was presented in Table 2.

Point 3. Is % TEM altered by treatment?

Response 3: We have checked that in patients with PD after chemotherapy the percentage of TEM was lower compared to the values before treatment. However, due to the small size of the study group (n = 3), we decided not to add it to the Results section.

Point 4. It would be useful to demonstrate a direct effect of CLL cells on Tie2 expression/IL-10/VEGF production in the monocyte fraction - e.g. Does co-culture (direct or transwell) of CLL cells with healthy (allogeneic) monocytes affect the expression of these molecules? Can this be replicated by purified Ang-2?

Response 4: Dear Reviewer, thank you very much for this valuable suggestion. We will seriously consider doing it in (hopefully) the near future. Unfortunately, we are unable to do it for the current revision. Due to the overall situation, there is currently scarcely any fresh material from CLL patients available to us. Therefore, we are unable to perform high-quality experiments.

Point 5. Do CLL TEM have lower levels of TNF and IL-12?

Response 5: Tie2+ and Tie2- monocytes were analysed for IL-12 expression by flow cytometry/RT-qPCR. However, detectable amounts of IL-12 protein/mRNA have been found in 5 out of 30 CLL samples. Moreover, the difference between Tie2+ and Tie2- fraction was not significant, so we decided not to add it to the Results section.

Minor comments

Point 1. Methods: How were the CD14+Tie2+/- fractions purified?

Response 1: Tie2-positive and Tie2-negative monocytes were sorted using BD FACSAria II flow cytometer. In the revised version of our manuscript, a piece of appropriate information has been added in the “Materials and Methods” section, point 2.4. “Tie2-positive and Tie2-negative monocytes sorting for RT-qPCR”, line:128-134.

Point 2. Results: What is the denominator for Tie2+ monocytes? PBMCs? CD19- cells? CD14+ cells?

Response 2: TEM were gated from total PBMC. The results are expressed as the percentage of CD14+CD16+ cells with Tie2 expression. (“Materials and Methods” section, point 2.2. “Flow cytometry analysis of TEM”, line: 114-115). In Figure 1 each dot plot shows the input gate in the title. Additionally, the percentage of cells for each gate was shown.

Point 3. Minor English language editing would improve the feel and flow of the manuscript

Response 3: We improved our English.

Reviewer 2 Report

The authors offers a descriptive study regarding the prognostic value of Tie2-expressing monocytes in CLL patients. The work partially extends from previously published work from Maffei et al (hematologica 2013), to include assessments on a large number of patients samples as well as clinical correlation. Overall well performed and statistically sound.

Author Response

Thank you very much for reviewing our work. We so appreciate you for this high evaluation of this manuscript.

Reviewer 3 Report

Review Manuscript #1195099:

Manuscript Summary:

            To clarify the immunosuppressive function of the circulating TEM (Tie-expressing monocytes) in Chronic Lymphocytic Leukemia (CLL), Wos J et al. studied the prognostic value of these cells in a large cohort of patients with their annotated biological parameters. After defining TEM on CD14, CD16 and Tie2 markers by flow cytometry, the authors showed an increased percentage of TEM in CLL samples compared to those observed in healthy volunteers. Based on statistical analysis, the authors showed that the larger number of TEM is linked to high risk disease (Rai stages), to adverse prognostic factors (Zap70, Del 11 and 17, IGHV mutational status) and clinical outcome from CLL patients (requiring treatments, responding to therapies, surviving to cures). After calculation of the optimal threshold for the percentage of TEM and using univariate and multivariate analyses, the authors demonstrated that higher TEM frequency is significantly associated with shorter time to treatment and overall survival from CLL patients. The authors also highlighted that CD14+/CD16+/Tie2+ cells (TEM) express higher levels of IL-10 (mRNA and protein), as well as more VEGF mRNA. They concluded that TEM are part of the CLL microenvironment that contribute to CLL progression and might be considered as a prognostic indicator.

            The manuscript is well written. The “Introduction” and “Materials and Methods” sections are short and appropriately referenced. The “Results” section is based on clear figures and the findings are quite convincing (see below). The “Discussion” paragraph clarifies the findings by putting them into the context of CLL and other diseases. Altogether, this manuscript makes contribution to the field and the major and minor concerns (please see below) should strengthen the data to the point where it may be acceptable.

Major concerns:

  • Line 202: In the title of Figure 1, the authors should precise the representative sample (CLL or healthy volunteer).

In this figure, authors should add the calculated percentage of cells for each dot plot shown (% of live cells, % of CD14, % of CD16…). For example, it looks like none of the PBMC are dead (panel c).

To better define the involved monocytes (classical, intermediate and non-classical monocyte subsets) in CLL and healthy PBMC, the authors should show the dot plot of the live cells that were double-stained for CD14-V450 and CD16-FITC (and not each labelling against SSC-A). Do the authors know whether the described TEM are inflammatory or patrolling monocytes?

  • Line 344 (Figure 7), do IL-10 and VEGF be expressed from specific monocyte subsets?

Minor concerns:

  • Figure 3: The police should be homogenized between panels;
  • Figure 4, panel b: The authors should explain why among the patients requiring therapy they put together CR and PR in the “responding group” and SD and PD in the “non-responding group”? It is difficult to consider that SD cases belong to the non-responding or responding groups. Why not to show each sub-group?
  • Figure 5: The ROC and AUC were evaluated to determine the most significant cutoff value of TEM percentage that distinguished Zap70+ and 70- cases. The authors should precise why they used Zap70 and not the other adverse prognostic factors, such as IGHV mutational status or genetic aberrations.
  • Figure 6 panels a and b: Y axis should be at 0.
  • Line 344: “… are shown in bold. .Only variables…” should be corrected.
  • Figures 7d, 8a and 8b: The authors should show graphs with linked points in order to determine whether the IL-10 expression (protein and mRNA) and VEGF expression are systematically increased between CD14+/CD16+/Tie2- and CD14+/CD16+/Tie2+. Did the authors observe high or low IL10/VEGF expressions in some CLL cases? 
  • Figure 8: What does mean 2-ΔCq? Is the value equal to 2-ΔCt?
  • Figures 7d and 8a: Is there a correlation between the IL10 protein and IL10 mRNA in Tie2- and Tie2+ cells of each case from the CLL cohort (n=30)?

Author Response

Dear Reviewer 3, we appreciate your comments. Indeed, in its past form, our manuscript suffered from several weaknesses. We carefully studied your suggestions and raised questions. We improved our work according to your remarks. We believe that this paper will meet your expectations. Please find the more detailed answers to your suggestions and questions below.

Major concerns:

Point 1. Line 202: In the title of Figure 1, the authors should precise the representative sample (CLL or healthy volunteer).

Response 1: The caption to Figure 1 has been modified.

 Point 2. In this figure, authors should add the calculated percentage of cells for each dot plot shown (% of live cells, % of CD14, % of CD16…). For example, it looks like none of the PBMC are dead (panel c).

Response 2: Figure 1 has been modified. The percentage of cells for each dot plot/gate was shown.

 Point 3. To better define the involved monocytes (classical, intermediate and non-classical monocyte subsets) in CLL and healthy PBMC, the authors should show the dot plot of the live cells that were double-stained for CD14-V450 and CD16-FITC (and not each labelling against SSC-A). Do the authors know whether the described TEM are inflammatory or patrolling monocytes?

Response 3: Figure 1 has been modified. Each dot plot shows the input gate in the title. Selected CD16-positive monocytes included “nonclassical” (CD14dimCD16+) and “intermediate” (CD14+CD16+) monocytes. Both groups together were analysed for Tie2 expression (Figure 1d). CD14+CD16- were “classical” monocytes. CD14dimCD16+ have e.g. high levels of the adhesion-related receptor CX3CR1 and exhibit an ability to patrol the vasculature. However, in our analysis, we did not perform this staining. The analysis included CD14+CD16+ cells without their differentiation into "non-classical" and "intermediate".

Point 4. Line 344 (Figure 7), do IL-10 and VEGF be expressed from specific monocyte subsets?

Response 4: CD14+CD16+Tie2+ (TEM) and CD14+CD16+Tie2- monocytes were analysed for IL-10 and VEGF expression. The exact point of adding the comparison between Tie2+ and Tie2- monocytes was to control which of those molecules are more specific for Tie2-positive monocytes and are probably the main perpetrators of their function. Figure 7 has been modified.

Minor concerns:

Point 1. Figure 3: The police should be homogenized between panels;

Response 1: The policy has been homogenized between panels.

Point 2. Figure 4, panel b: The authors should explain why among the patients requiring therapy they put together CR and PR in the “responding group” and SD and PD in the “non-responding group”? It is difficult to consider that SD cases belong to the non-responding or responding groups. Why not to show each sub-group?

Response 2: Figure 4b has been modified. Each sub-group (CR, PR, SD, and PD) was shown. The results (line 263-267) has been modified.

Point 3. Figure 5: The ROC and AUC were evaluated to determine the most significant cutoff value of TEM percentage that distinguished Zap70+ and 70- cases. The authors should precise why they used Zap70 and not the other adverse prognostic factors, such as IGHV mutational status or genetic aberrations.

Response 3: ZAP-70 was used in ROC curve analysis because in our previous studies ZAP-70 has been shown as one of the most powerful prognostic factors. Basing on the ROC curve, we determined the best threshold for the percentage of TEM that was associated with ZAP-70 above 20%. This was pointed in the "Statistical analysis" (line: 189-192). IGHV mutation status was not used in the ROC and AUC analysis. Unfortunately, we have limited data about IGHV mutational status. It was accessible only for 64 participants of this study.

Point 4. Figure 6 panels a and b: Y axis should be at 0.

Response 4: We have modified Figure 6 panels a and b. Y-axis is at 0.

Point 5. Line 344: “… are shown in bold. .Only variables…” should be corrected.

Response 5: This line has been modified. The sentence in current form:All statistically significant variables (p £ 0.05), as found in the univariate analyses, were included in multivariate analysis based on a Cox proportional hazards model

 Point 6. Figures 7d, 8a and 8b: The authors should show graphs with linked points in order to determine whether the IL-10 expression (protein and mRNA) and VEGF expression are systematically increased between CD14+/CD16+/Tie2- and CD14+/CD16+/Tie2+. Did the authors observe high or low IL10/VEGF expressions in some CLL cases?

Response 6: Figures 7d, 8a and 8b have been modified. As suggested a graph with linked points was used. We can more clearly see the rate of slope between individual data points. No association was observed between IL-10 / VEGF expression and specific CLL cases. Probably a larger cohort of patients is necessary.

Point 7. Figure 8: What does mean 2-ΔCq? Is the value equal to 2-ΔCt?

Response 7: Threshold cycle (Ct) has been given multiple names over the years including, crossing point (Cp), take-off point (TOP) and quantification cycle (Cq). These values are all the same, just with different names. Throughout our manuscript, we used Cq only. In our study, data were normalized to β-actin expression (endogenous control) and presented as 2-ΔCq. ΔCq is the difference between the cycle quantification value (Cq) of the target gene (Cqt) and the reference gene (Cqr) (ΔCq = Cqt - Cqr). We added this information into a revised version of our manuscript (“Materials and Methods” section; part 2.5. “RT-qPCR for IL-10, and VEGF”; line 144-145)

Point 8. Figures 7d and 8a: Is there a correlation between the IL10 protein and IL10 mRNA in Tie2- and Tie2+ cells of each case from the CLL cohort (n=30)?

Response 8: There was a positive correlation between the IL-10 protein and IL-10 mRNA expression in Tie2+ (r = 0.781; p <0.01) and Tie2- (r = 0.538; p < 0.01) monocytes. In the revised version of our manuscript, we added information about mentioned correlation (line 

Round 2

Reviewer 1 Report

I don't feel that they have satisfactorily addressed the following points:   Point 1 - the authors have stated that they will address this point in their future work. Why not as part of this research?   Point 3 - They should expand the number of patients (currently n = 3) to address this   Point 4 - Again they defer this to "future" work and cite lack of availability to patient samples due to the current situation - presumably they are referring to Covid-19. This point is valuable to implicate a direct effect of the CLL cells on TEM and these experiments can be performed using cryopreserved CLL cells. I am not aware of journals generally accepting the Covid-19 pandemic as justification for lack of experimental work. 

Author Response

Dear Reviewer 1, we appreciate your comments. Indeed, in its past form, our manuscript suffered from several weaknesses. We improved our work according to your remarks. We believe that this paper will meet your expectations. Please find the more detailed answers to your suggestions below.

Point 1. Was TEM high/low status predictive of response to therapy?

Response 1: We have performed a study aimed to examine the significance of TEM percentage in predicting treatment outcome. This analysis was added in “Results” (line: 379-389) and „Discussion" (line: 507-510). Overall response rate (ORR) and time to re-treatment (TTR) were included in the analysis. However, no statistically significant differences were noticed between the TEMhigh and TEMlow group.

Point 3. Is % TEM altered by treatment?

Response 3: Ten patients requiring treatment were examined at two time-points: at the time of diagnosis and before the start of the treatment. In addition, 9 patients were examined at 6 or 12 months after chemotherapy. One patient died during the observation period. We added this issue into a revised version of our manuscript (“Results” line 303-304, 307-309 and “Discussion” line: 496-499). A column "after treatment" was added to Table 2. We noted that the percentage of TEM in individual patients changed over time and was significantly higher just before the initiation of chemotherapy than at the time of diagnosis or after treatment.

Point 4. It would be useful to demonstrate a direct effect of CLL cells on Tie2 expression/IL-10/VEGF production in the monocyte fraction - e.g. Does co-culture (direct or transwell) of CLL cells with healthy (allogeneic) monocytes affect the expression of these molecules? Can this be replicated by purified Ang-2?

Response 4: Monocytes from healthy donors were cultured in direct contact with CLL cells. Cryopreserved PBMC from six CLL patients were used. Healthy monocytes were isolated from freshly PBMCs. We have added this to the “Materials and Methods” section, point 2.6. “Coculture conditions” (Line:  147-171). We also added this study into ”Results”, point 3.7. “The direct effect of CLL cells on IL-10 or VEGF expression in healthy monocytes” (line 429-441) and "Discussion"(line 542-547). Unfortunately, we did not have a purified Ang-2. Therefore, it was not used in culture.  New Figure 9 was also added (line 442-450).

Reviewer 3 Report

cancers-1195099-peer-review-v2

I would like to thank the authors for their modified manuscript in response to the comments I wrote and questions I asked. 

To my opinion, the manuscript should be accepted after few corrections in the text:

  1. Introduction -line 54- "Although" should be changed by "However";
  2. Materials and Methods - line 94- What does mean "interphase" cells in the new sentence?
  3. Materials and Methods - line 122- 20 min. should be corrected by removing the dot;
  4. Materials and Methods - line 124- "After that" should be changed by “Then”; 
  5. Results - line 200- TEMs should be replaced by TEM;
  6. Results - line 208- "... from the CLL case" should be modified by "... from a CLL case";
  7. Results - Figures 2 (line 234), 3 (line 251) and 4 (line 274)- The Y axis of each panel should be modified by adding TEM (Tie+ monocytes) %; 
  8. Results - line 238- TEM percentage and CLL adverse prognostic factors;
  9. Results - line 286- Comparison of CLL TEMhigh and TEMlow patient groups;
  10. Results - line 315- The title of the Paragraph 3.5 should be changed and summarize the main result;
  11. Results - Figure 6 -line 340- Panel a, Y axis, Cumulative proportion surviving;
  12. Results - Title -line 350- title should be changed by "IL-10 and VEGF are overexpressed by CLL TEM";
  13. Results - line 367/369- The added sentence should be moved after the next sentence (after "... Figure 8a)" and before ". Moreover" line 371);
  14. Results - line 374- The authors should add a sentence for giving the rational of why they measured the Ang-2 concentration in peripheral blood plasma of CLL patients. To improve the manuscript, the authors should add the result of the correlation between Ang-2 concentration and VEGF mRNA (as discussed in lines 448 to 464). 

Author Response

Dear Reviewer 3, thank you very much for your valuable suggestions and comments. We improved our work according to your remarks. Please find the more detailed answers to your suggestions below.

Point 1. Introduction -line 54- "Although" should be changed by "However";

Response 1. "Although" was changed by "However"

Point 2. Materials and Methods - line 94- What does mean "interphase" cells in the new sentence?

Response 2. We have rephrased „ After washing twice interphase cells were resuspended in phosphate-buffered saline (PBS)„ into  “Mononuclear cells at the interphase were removed, washed twice and resuspended in phosphate-buffered saline (PBS)” (line: 93-94)

Point 3. Materials and Methods - line 122- 20 min. should be corrected by removing the dot;

Response 3. The dot was removed.

Point 4. Materials and Methods - line 124- "After that" should be changed by “Then”;

Response 4. “After that" was changed by “Then”

Point 5. Results - line 200- TEMs should be replaced by TEM;

Response 5. TEMs was replaced by TEM  (now line 225)

Point 6. Results - line 208- "... from the CLL case" should be modified by "... from a CLL case";

Response 6. The article was removed (now line 233)

Point 7. Results - Figures 2 (line 234), 3 (line 251) and 4 (line 274)- The Y axis of each panel should be modified by adding TEM (Tie+ monocytes) %; 

Response 7. The Y axis of each panel was modified. “Tie+ monocytes (%)” was changed by “TEM (Tie+ monocytes) %”.

Point 8. Results - line 238- TEM percentage and CLL adverse prognostic factors;

Response 8. “CLL” was added (now line 263).

Point 9. Results - line 286- Comparison of CLL TEMhighand TEMlow patient groups;

Response 9. “CLL” was added (now line 315).

Point 10. Results - line 315- The title of the Paragraph 3.5 should be changed and summarize the main result;

Response 10. The title of paragraph 3.5 was changed to “The high percentage is associated with a shorter time to treatment and poor overall survival” (now line 344).

Point 11. Results - Figure 6 -line 340- Panel a, Y axis, Cumulative proportion surviving;

Response 11. The spelling was corrected (line 370).

Point 12. Results - Title -line 350- title should be changed by "IL-10 and VEGF are overexpressed by CLL TEM";

Response 12. “CLL” was added  (now line 390)

Point 13. Results - line 367/369- The added sentence should be moved after the next sentence (after "... Figure 8a)" and before ". Moreover" line 371);

Response 13. The sentence “ was moved after Figure 8b (line 409-411)

Point 14. Results - line 374- The authors should add a sentence for giving the rational of why they measured the Ang-2 concentration in peripheral blood plasma of CLL patients. To improve the manuscript, the authors should add the result of the correlation between Ang-2 concentration and VEGF mRNA (as discussed in lines 448 to 464).

Response 14. A sentence “The number and activity of TEM are probably directly dependent on Ang-2 [7,10]. Therefore, we decided to assess its concentration in the plasma of CLL patients.” Was added (line: 414-415 ). In a revised version of our manuscript, we added the correlation between Ang-2 concentration and VEGF mRNA was added (line: 420-422).
